# Engineered natural killer cells impede the immunometabolic CD73-adenosine axis in solid tumors

Andrea M Chambers[1], Kyle B Lupo[1], Jiao Wang[1], Jingming Cao[1], Sagar Utturkar[2], Nadia Lanman[2,3], Victor Bernal-Crespo[4], Shadia Jalal[5], Sharon R Pine[6,7,8], Sandra Torregrosa-Allen[2], Bennett D Elzey[2], Sandro Matosevic[1,2]*

[1]Department of Industrial and Physical Pharmacy, Purdue University West Lafayette, West Lafayette, United States; [2]Center for Cancer Research, Purdue University West Lafayette, West Lafayette, United States; [3]Department of Comparative Pathobiology, Purdue University, West Lafayette, United States; [4]Histology Research Laboratory, Center for Comparative Translational Research, College of Veterinary Medicine, Purdue University, West Lafayette, United States; [5]Department of Medicine, Division of Hematology/Oncology, Indiana University School of Medicine, Indianapolis, United States; [6]Rutgers Cancer Institute of New Jersey, Rutgers, The State University of New Jersey, New Brunswick, United States; [7]Department of Pharmacology, Robert Wood Johnson Medical School, Rutgers, The State University of New Jersey, New Brunswick, United States; [8]Department of Medicine, Robert Wood Johnson Medical School, Rutgers, The State University of New Jersey, New Brunswick, United States

*For correspondence:
sandro@purdue.edu

**Competing interest:** The authors declare that no competing interests exist.

**Abstract** Immunometabolic reprogramming due to adenosine produced by CD73 (encoded by the 5′-ectonucleotidase gene *NT5E*) is a recognized immunosuppressive mechanism contributing to immune evasion in solid tumors. Adenosine is not only known to contribute to tumor progression, but it has specific roles in driving dysfunction of immune cells, including natural killer (NK) cells. Here, we engineered human NK cells to directly target the CD73-adenosine axis by blocking the enzymatic activity of CD73. In doing so, the engineered NK cells not only impaired adenosinergic metabolism driven by the hypoxic uptake of ATP by cancer cells in a model of non-small-cell lung cancer, but also mediated killing of tumor cells due to the specific recognition of overexpressed CD73. This resulted in a 'single agent' immunotherapy that combines antibody specificity, blockade of purinergic signaling, and killing of targets mediated by NK cells. We also showed that CD73-targeted NK cells are potent in vivo and result in tumor arrest, while promoting NK cell infiltration into CD73+ tumors and enhancing intratumoral activation.

## Editor's evaluation

The adoptive transfer of NK cells expressing chimeric antigen receptors (CARs) has generated significant interest for cancer immunotherapy, considering recent success of CAR-NK cells in treating B cell malignancies with fewer deleterious side effects, as compared to CAR-T cell therapies. In this paper, the authors engineered NK cells expressing a CAR targeting CD73, which is highly expressed in lung adenocarcinomas and generates immunosuppressive adenosine. Their CD73.CAR-NK cells not only exhibited improved infiltration of and enhanced cytotoxicity toward CD73-expressing tumor cells, but also significantly reduced adenosine production by a lung adenocarcinoma line, which has potential to further improve anti-tumor immune responses.

## Introduction

Lung cancer is the leading cause of cancer-related death worldwide and in the United States. The overall 5-year survival rate is also very low for both men and women *Sidney et al., 2019*; *Howlader et al., 2014*. Approximately 85% of all lung cancers diagnosed are non-small cell lung cancers (NSCLC), and almost 40% of those cases are lung adenocarcinomas *Zappa and Mousa, 2016*. Standard of care therapy has evolved to include immunotherapy as a first-line treatment for early and more advanced stages of lung cancer, and typically these treatments are checkpoint inhibitors, focusing on PD-1/PD-L1 blockade *Socinski et al., 2018*; *Huang et al., 2017* alone or in combination with chemotherapy. For lung cancer patients without a driver mutation, immunotherapy has resulted in remarkable responses in a subset of NSCLC patients. *Gandhi et al., 2018* However, even with a combination of chemotherapy and checkpoint inhibitor immunotherapy, only 34% of patients are alive and progression-free after 1 year.

Another limitation of currently approved immunotherapies in NSCLC is that patients with targetable mutations do not seem to respond to these agents. Patients with epidermal growth factor receptor (EGFR) or anaplastic lymphoma receptor tyrosine kinase (ALK) abnormalities, for instance, do not qualify for PD-1/PD-L1 treatments due to low response rates. Even though these patients still have a number of targeted therapy options besides immunotherapy, the cancer can eventually progress, even with commonly used tyrosine kinase inhibitors *Liu et al., 2020*; *Xia et al., 2019*; *Yoneshima et al., 2018*. Additionally, patients with non-targetable genotypes (for instance, some non-targetable KRAS mutant/STK11 mutant tumors) do not respond well to standard treatment either *Sidney et al., 2019*. Poor efficacy to PD-1/PD-L1 in these patients may be due to a number of factors including a decrease in certain tumor-infiltrating lymphocytes and an increase in infiltration of regulatory T cells and immunosuppression in EGFR mutated tumors *Xia et al., 2019*; *Yu et al., 2018*; *Sugiyama et al., 2020*; *Velez and Burns, 2019*; *Streicher et al., 2017*; *Lin et al., 2019*.

In the tumor microenvironment (TME), altered metabolism in the tumor cells due to hypoxia, nutrient deficiency, and increased glucose metabolism can lead to modified immune function and tumorigenesis due to the reorganization of the immune cells' metabolism *Annibaldi and Widmann, 2010*. In this hypoxic environment, high levels of ATP are released from the cancer cells that are then consecutively degraded through ectoenzymes CD39 and CD73 to become adenosine (ADO) *Wang and Matosevic, 2018*; *Poth et al., 2013*; *Leone and Emens, 2018*; *Antonioli et al., 2016*. Adenosine can then accumulate in the TME and severely impede immune cell function and activity *Wang and Matosevic, 2018*; *Chambers et al., 2018*. CD73 was shown to be a negative prognostic factor for patient survival in non-small cell lung cancer *Inoue et al., 2017*. In EGFR-mutated lung adenocarcinoma, CD73 becomes highly over-expressed and has been shown to have less infiltrating CD4[+] T cells compared to non-mutated NSCLC *Velez and Burns, 2019*; *Streicher et al., 2017*. This overexpression of CD73 in mutated EGFR, for instance, may provide one explanation to the poor responses seen with current immunotherapy options. Therefore, there is an urgent need to develop better strategies to overcome checkpoint inhibitor resistance in both mutated and wildtype lung adenocarcinoma.

Chimeric antigen receptor (CAR) T-cell therapy has shown to be a promising approach preventing tumor progression; however, CAR T-cells have only been successful in treating blood hematologic malignancies. Also, autologous CAR T-cells have been shown to cause cytokine storm and induce graft-versus host disease (GvHD). CAR NK-cells, on the other hand, may induce minimal side effects and allow the ability to use allogeneic cells, paving the way for an 'off the shelf' immunotherapeutic treatment *Yilmaz et al., 2020*; *Siegler et al., 2018*; *Lupo and Matosevic, 2019*. In addition, most CARs are derived from T-cell receptor intracellular signaling domains *Siegler et al., 2018*; therefore, the use of an NK stimulatory domain may help to increase NK cell cytotoxicity in the tumor microenvironment.

Here, we demonstrate the use of an engineered peripheral blood NK cell-based therapy for overcoming the immunosuppression induced by cancer-producing adenosine. We have engineered the NK cells to directly target the CD73 on the tumor cells by imparting a CD73-specific CAR along with NK-specific signaling derived from FCγRIIIa (CD16). We report that these anti-CD73 CAR-NK cells have substantial anti-tumor activity and enhanced degranulation and cytokine production in vitro as well as halted tumor growth in vivo against CD73[+] lung adenocarcinoma. This study demonstrates the potential of CD73 as an immunotherapeutic target for solid tumors, by enabling tumor-specific recognition and targeting of metabolic reprogramming by directly impairing the CD73-adenosine axis.

## Results

### Generation of CD73-targeted NK cells

CD73 is abundantly expressed in NSCLC, specifically lung adenocarcinoma (*Figure 1A*), where it is a recognized prognostic marker associated with poorer overall survival (OS) *Inoue et al., 2017* (*Figure 1B*). Although it is also expressed in small-cell lung cancer (SCLC, Figure S2 – *Figure 2—figure supplement 1*), its expression does not differ between males *vs* females in either NSCLC or SCLC. (*Figure S2 – Figure 2—figure supplement 1*). The expression of CD73 causes the accumulation of adenosine, a metabolite that we and others have shown to interfere with NK cell cytotoxic ability, proliferation, and trafficking activities, through interaction with A2A receptors on the NK cell surface *Chambers et al., 2018*. Briefly, CD73 catalyzes the accumulation of extracellular ADO from AMP, at levels higher than normal tissue, which stimulates the A2A adenosine receptors on the NK cell surface to inhibit their proliferative and cytolytic functions. Expression of CD73 is also high and localized on tumor cells in lung adenocarcinoma KP PDX models (*Figure 1C*). GSEA analysis of TCGA patient data indicates that adenosine interferes with pathways involved in NK cell activation, migration and cytotoxic activities in lung adenocarcinoma (*Figure 1D*, *Figure 1—figure supplements 1–4*), with inhibition of NK cell cytotoxicity showing the most significant negative correlation with CD73[+] lung adenocarcinoma.

To prevent the buildup of adenosine due to the enzymatic activity of CD73, and thus avoid the risk of it subsequently inducing suppression of NK cell effector responses in the tumor microenvironment, we sought to investigate the potential of directly targeting CD73 using NK cells. We developed a genetic construct to enable targeting of CD73 and neutralization of its enzymatic activity with NK cells. To do so, we fused a functional, neutralizing CD73 scFv with intracellular and transmembrane regions of FC(gamma)RIIIa (CD16) (*Figure 1E*). As a first step toward functional evaluation, we were interested in evaluating the efficiency of mRNA transfection and viral transduction in expressing the CD73-directed construct on NK cells. Plasmids encoding for human CD73 scFv and FCγRIIIa (CD16) regions were developed using the pcDNA3.1(+) for mRNA transfection and the lentiviral pLV plasmid for lentiviral transduction. Although mRNA is transient and generally results in lower transduction efficiencies with primary human NK cells, electroporation was used to validate the function of the gene with mRNA before transduction with the lentivirus.

To determine the optimal electroporation conditions for inserting the CD73 scFv CAR into peripheral blood NK cells, a variety of conditions were tested, ranging 250 V-500V with various pulse durations, pulse lengths, and pulse intervals (Figure S6: *Figure 2—figure supplement 3*). The conditions of 300 V, 4ms, 5 pulses, with a 5-s pulse interval was determined to result in the highest gene expression we achieved using electroporation on human NK cells (*Figure 2A*). The expression of the anti-CD73 scFv showed that an average of 25% of NK cells were found to express the CD73-CAR construct (*Figure 2A*). Since the loss of NK cell viability is commonly associated with electroporation, we evaluated the viability of NK cells in culture after transfection. Under optimal electroporation conditions, we did not observe any decrease in cell viability due to the mRNA electroporation procedure (*Figure 2B*).

We next evaluated whether transduction could improve expression of the CD73-CAR. Primary NK cells are known to be difficult to transduce *Gong et al., 2020*; *Nanbakhsh et al., 2018*; *Waller, 2017*; *Müller et al., 2019*; *Micucci et al., 2006*, and to identify optimal conditions for the transduction of primary NK cells, a variety of conditions were tested (Figure S7 – *Figure 2—figure supplement 3*). We determined that the optimal condition for transduction was obtained using a spinoculation protocol with protamine sulfate (10 μg/mL) and IL-12 (1.5 ng/mL).

For lentivirally transduced NK cells, enrichment of CD73-CAR[+] NK cells after sorting following transduction yielded close to 100% purity of the lentivirally-transduced CD73.CAR-NK cell population of primary NK cells (*Figure 2C*). With transduced NK cells, moreover, there was no donor-specific drop in viability and no detrimental effect of cell growth of the donor NK cells due to the transduction procedure, unlike what we had observed with electroporation. In fact, the transduced NK cells had slightly higher cell growth compared to non-transduced cells (*Figure 2D*). The transduced NK cell population also was not altered in its proportion of CD56[+] NK cells compared to non-transduced NK cells (*Figure 2E*), and they could expand in media to reach clinical doses. Although there can be some variability in gene expression on NK cells due to donor variability, overall high levels of expression (*Figure 2C*) of the CD73 CAR with no changes in NK cell phenotypic populations were obtained. In summary, we showed that lentiviral transduction yields superior expression of the CD73.CAR on NK

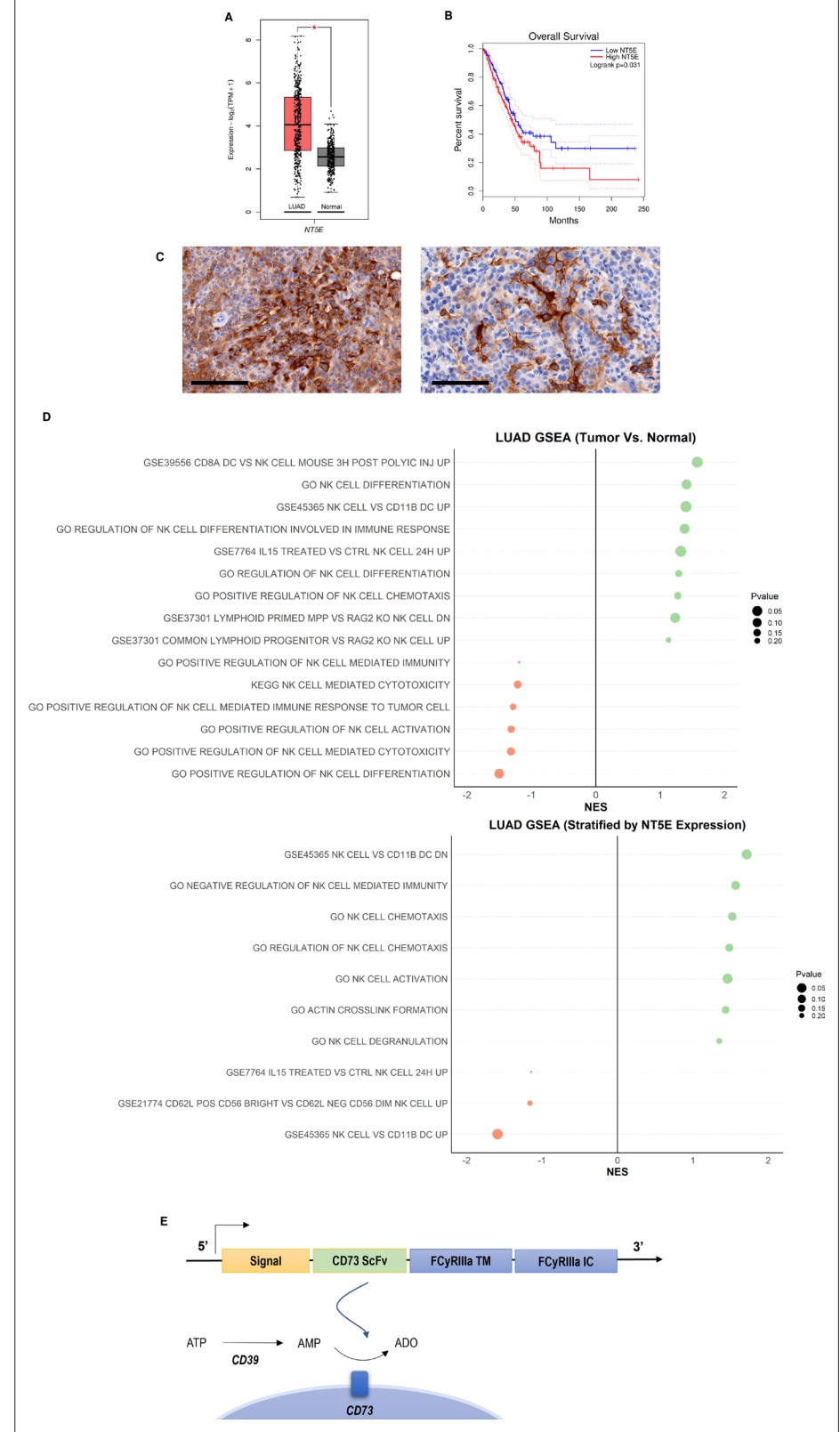

**Figure 1.** CD73 is overexpressed in lung cancer and is associated with poor overall survival and impairment of NK cell functions. (**A**) Expression of *NT5E* (CD73) in lung adenocarcinoma (N=483) and normal lung (N=387) based on analysis of TCGA patient data. (**B**) Kaplan-Meier plots of overall survival of lung adenocarcinoma patients (N=478) patients from TCGA dataset via GEPIA2 based on expression of *NT5E* as a prognostic marker for overall survival.

*Figure 1 continued on next page*

*Figure 1 continued*

Hazards ratio = 1.4; 95% confidence interval. (**C**) (*Left*) IHC staining for CD73 of intratracheal K-ras$^{LSL-G12D/+}$;p53$^{fl/fl}$ (KP) tumor in LSL-KrasG12D/Tp53flox/flox mice on day 22 (20×); scale bar = 100 μm; (*Right*) CD73 IHC staining for CD73 on subcutaneous KP tumors in on day 31 (20×). (**D**) GSEA chart of significant pathways (pvalue <0.25) related to natural killer cells in TCGA-LUAD patients (*top*) comparison between tumor and normal samples (*bottom*) comparison between high and low groups stratified by *NT5E* gene expression. (**E**) Structure of the CD73. CAR construct consisting of a CD73 scFv, and transmembrane and intracellular domains of FCyRIIa. The CD73 scFv recognizes CD73 to block its enzymatic activity and reduce the accumulation of extracellular adenosine. Volcano plots and GSEA data for LUAD TCGA RNASeq analyses are shown in Figures S2-S5 (*Figure 1—figure supplements 1–4*). Raw NES and pvalues for TCGA analyses are provided in Tables S1 and S2 (*Figure 1—source data 1*, *Figure 1—source data 2*).

The online version of this article includes the following source data and figure supplement(s) for figure 1:

**Source data 1.** Raw NES and p values for TCGA analyses.

**Source data 2.** Raw NES and p values for significant pathways stratified for *NT5E*.

**Figure supplement 1.** Bioinformatic analysis of expression of CD73 (*NT5E*) in NSCLC and SCLC.

**Figure supplement 2.** Volcano plot of differential expression genes (tumor vs normal) in lung adenocarcinoma based on bioinformatics analysis of TCGA patient data.

**Figure supplement 3.** Volcano plot of differential expression genes (tumor vs normal) in *NT5E*-expressing lung adenocarcinoma based on bioinformatics analysis of TCGA patient data.

**Figure supplement 4.** Gene set enrichment analysis of lung adenocarcinoma TCGA data related to natural killer cells.

cells with conditions that can eliminate large losses in viability and that can generate a pure population of CAR-expressing cells.

## NK cells engineered with a CD73.CAR target CD73$^+$ lung adenocarcinoma in both normoxic and chemically induced hypoxic conditions, while not impacting normal cells

Compared to control primary NK cells, the CD73.CAR NK cells engineered via either the pcDNA3.1(+) (CD73.mRNACAR) or pLV (CD73.CAR) plasmids exhibited significantly higher killing of A549 cells after 4 hr at various effector:target (E:T, PNK:A549) ratios (*Figure 3A–B*). Killing abilities of primary NK cells were generally highly dependent on donor NK cells, precluding direct comparison of killing efficacies. Optimal E:T killing ratios were also different between different donors, both for mRNA-electroporated as well as lentivirally transduced NK cells. PNK cells transduced with lentivirus, compared with mRNA electroporated PNK cells, had a greater relative increase in killing ability between engineered and non-engineered cells, which may be due to higher gene expression and non-transient expression obtained using lentiviral transduction. Conversely, lower killing of CD73 knockout A549 (KO.A549, *Figure S8*) cells was observed by CD73.CAR-NK cells compared to control, non-transduced primary NK cells, showing that mere expression of the CAR construct does increase the killing ability of human NK cells (*Figure 3B*). Therefore, we speculate that the extent of the in vitro killing ability of target cells by CD73.mRNACAR-NK cells is reliant on the enzymatic activity of CD73 on the target cells, which we observe may have limited adenosine production under in vitro conditions. Furthermore, co-culture with A549 cells stimulated the CAR-NK cells to induce increased degranulation displayed by cell surface CD107a expression (*Figure 3C and D*) and cytokine secretion shown by IFN-γ (*Figure 3D*). Generally, lentiviral transduction induced higher increases in degranulation and cytokine production by human NK cells between non-engineered and engineered cells compared to electroporation. In fact, IFN-γ production by electroporated NK cells was comparable to that by non-engineered NK cells, while degranulation only increased at one E:T ratio.

In the tumor microenvironment, hypoxia can negatively affect the ability of the NK cells to target and kill the tumor targets (*Figure 4A*). Hypoxia is also a known enhancer of CD73 activity via HIF-1α *Chambers and Matosevic, 2019*, leading ultimately to upregulation of adenosine production. However, the antitumor and adenosine blocking ability of the anti-CD73 CAR outperformed the non-engineered primary NK cells against A549 cells, especially at higher effector:target ratios under chemically induced hypoxia (*Figure 4B*).

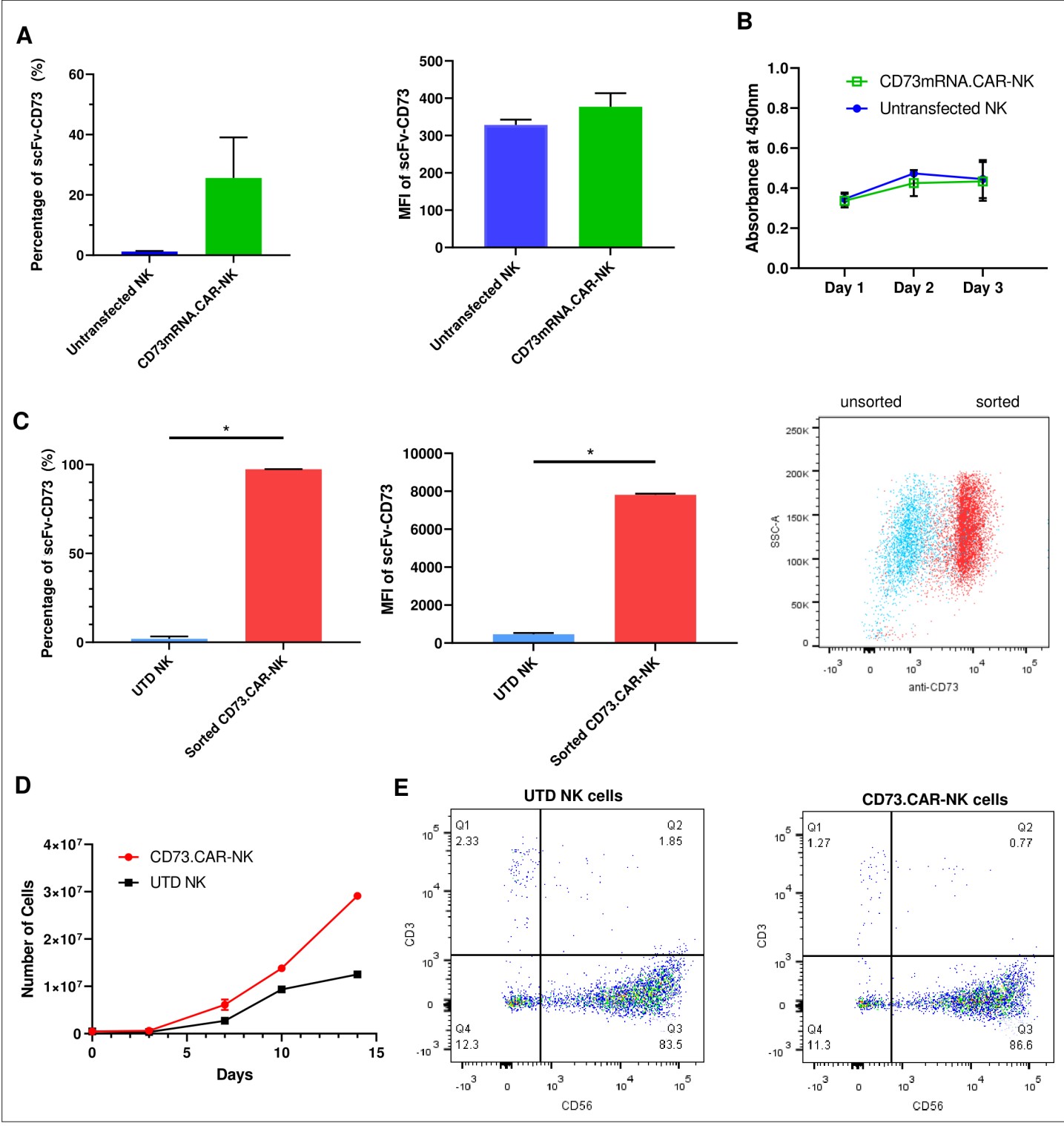

**Figure 2.** Engineering of CD73.CAR NK cells. (**A**) Transfection efficiency of CD73-mRNACAR construct in terms of (*left*) percentage and (*right*) MFI of the CD73-mRNACAR expressed by primary human NK cells following mRNA electroporation (n=3 samples from 3 individual donors). (**B**) Viability of CD73.mRNACAR-NK cells and non-transduced NK cells (NK) in culture following mRNA electroporation (n=3 samples). (**C**) Enrichment of lentivirally transduced CD73.CAR-NK cells following sorting in terms of (*left*) percentage and (*middle*) MFI of the CD73.CAR expressed by primary human NK cells following lentiviral transduction; (*right*) Dot plot of sorted populations of CD73.CAR[+] and CD73.CAR[−] transduced human NK cells (n=3 samples) (**D**) Expansion of CD73.CAR-NK cells and untransduced NK cells following lentiviral transduction (n=3 samples) (**E**) Proportion of CD56[+]CD3[−] untransduced human NK cells and CD73.CAR-NK cells following lentiviral transduction. *p<0.05, UTD = untransduced. See *Figure S1* (***Figure 2—figure supplement***

*Figure 2 continued on next page*

*Figure 2 continued*

*1*) for gating strategy for human NK cell isolation. See *Figure S6* (*Figure 2—figure supplement 2*) for data for transfection optimization studies. See *Figure S7* (*Figure 2—figure supplement 3*) for data for lentiviral transduction optimization studies.

The online version of this article includes the following figure supplement(s) for figure 2:

**Figure supplement 1.** Gating strategy for human NK cell identification by flow cytometry.

**Figure supplement 2.** Optimization of mRNA electroporation of human NK cells.

**Figure supplement 3.** Optimization of lentiviral transduction of human NK cells to express the CD73.CAR construct.

The killing ability of NK cells both electroporated with the CD73.mRNACAR [CD73-Fc(gamma)RIII_pcDNA3.1(+)] construct and lentivirally-transduced with the CD73.CAR construct were also compared to the efficacy of an anti-CD73 antibody (*Figure 4C and D*). Here, the anti-CD73 antibody performed similarly in vitro in terms of killing ability compared to the engineered CAR-NK cells, except at the higher ratio, where the CAR-NK cells outperformed the CD73 antibody. The engineered cells may be superior to the antibody at the higher ratios due to the decrease in antibody potency with higher amounts of NK cells. In addition, the NK cells did not experience any decrease in cytotoxic activity due to the expression of the CAR gene. By using an antibody with known CD73 neutralizing activity, this also confirms the extent of the ability of CD73 to be targetable in vitro.

In order to address potential off-target effects due to targeting of non-tumor healthy cells that express CD73, we evaluated the ability of the engineered NK cells to target CD73+ healthy endothelial cells. We chose HUVEC cells as they express high levels of CD73 (*Figure 4E*), and the CD73.CAR-NK cells may come in contact with endothelial cells when circulating throughout the body if infused systemically. Here, we showed that the CD73.mRNACAR-NK cells did not preferentially kill normal cells and the primary NK cells in general had very low killing against the healthy endothelial cells. The resistance to killing has been attributed to a number of mechanisms, including the protective role of ATP to NK cells, and HLA-E expression[73]. This suggests that CD73 recognition may not be the only mechanism involved in the ablation of target cells by CD73-targeting NK cells. The ability of the engineered NK cells to block adenosine production was also evaluated. As shown, the CD73.CAR-NK cells decreased adenosine production below baseline levels of A549 cells cultured alone (*Figure 4F*).

## Engineered NK cells efficiently target and infiltrate CD73+ lung adenocarcinoma tumors in vivo

To evaluate the in vivo antitumor activity of the anti-CD73 engineered NK cells, we established a subcutaneous xenograft model using lung adenocarcinoma CD73+ A549 cells engrafted into NRG mice (*Figure 5A*). To determine the tumor volumes, we used luciferase-tagged A549 cells so that we could accurately measure the tumor size both inside and on the surface of the mouse. Compared to both control PBS mice and primary non-engineered NK cells, the CD73.CAR-NK cells had a much more potent antitumor response and resulted in reduced CD73+ tumor growth. In addition, CD73.CAR-NK cells resulted in tumor growth arrest overtime, almost 50 days after tumor implantation, compared to the other treatment groups (*Figure 5B*). Mice treated with the CD73.CAR-NK cells also survived longer than mice in other treatment groups (*Figure 5C*). There was also no significant decrease in body weight of the mice in any of the groups throughout and after the treatment period (*Figure 5D*).

NK cell infiltration into the A549 tumors was detected via immunohistochemistry (IHC) staining following adoptive transfer therapy (*Figure 5E*). More NK cells were observed to infiltrate tumors treated with the CD73.CAR-engineered variants (*Figure 5F*). We detected significantly higher levels of granzyme B in tumors treated with CD73.CAR-NK cells, suggesting that NK cells that infiltrated those tumors were more cytotoxically active (*Figure 5G*). Additionally, activating markers DNAM-1, NKp30, and NKG2D (*Figure 5H*) and inhibitory markers LAG-3 and PD-1 (*Figure 5H*) were measured on circulating NK cells. We observed that there were no significant differences in both activating and inhibitory expression on engineered and non-engineered primary NK cells in circulation (*Figure 5H*).

Altogether, these studies demonstrate the first proof-of-principle of the cytotoxic and specific activity of CD73-targeted CAR-NK cells against lung adenocarcinoma. These responses also accompanied increased degranulation and cytokine release as well as NK cell infiltration and a striking arrest

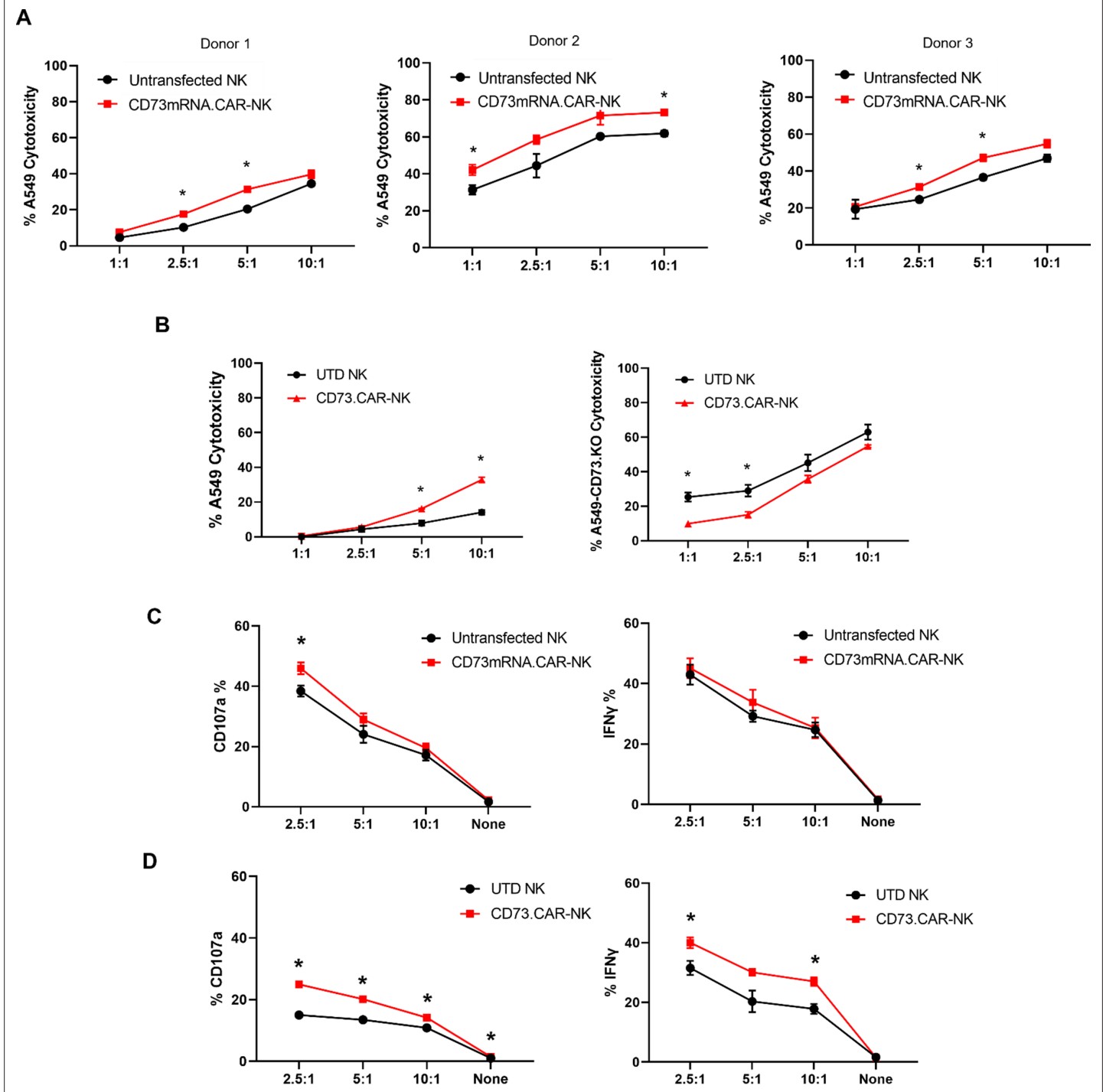

**Figure 3.** Functionality of CD73. CAR NK cells. (**A**) Donor-specific cytotoxicity of CD73.mRNACAR-NK cells and untransduced primary human NK cells against lung adenocarcinoma targets for three individual donors. (**B**) Cytotoxicity of CD73.CAR-NK cells and human NK cells against (*left*) WT lung adenocarcinoma targets (*right*) CRISPR/Cas9 CD73 KO lung adenocarcinoma cells (**C**) Degranulation, measured through CD107a expression, of CD73. mRNACAR-NK cells and human NK cells in response to lung adenocarcinoma targets. and IFN-γ production by CD73.mRNACAR-NK cells and human NK cells in response to lung adenocarcinoma targets. (**D**) CD107a and IFN-γ production by CD73.CAR-NK cells and human NK cells in response to lung adenocarcinoma targets. Data are shown representative of three donors, unless otherwise indicated. *p<0.05, UTD = untransduced. See *Figure S8* (*Figure 3—figure supplement 1*) for expression histograms of *NT5E* on WT and CD73KO A549 cells.

The online version of this article includes the following figure supplement(s) for figure 3:

**Figure supplement 1.** Histograms for expression of CD73 on (*top*) WT and (*bottom row*) CD73KO A549 cells.

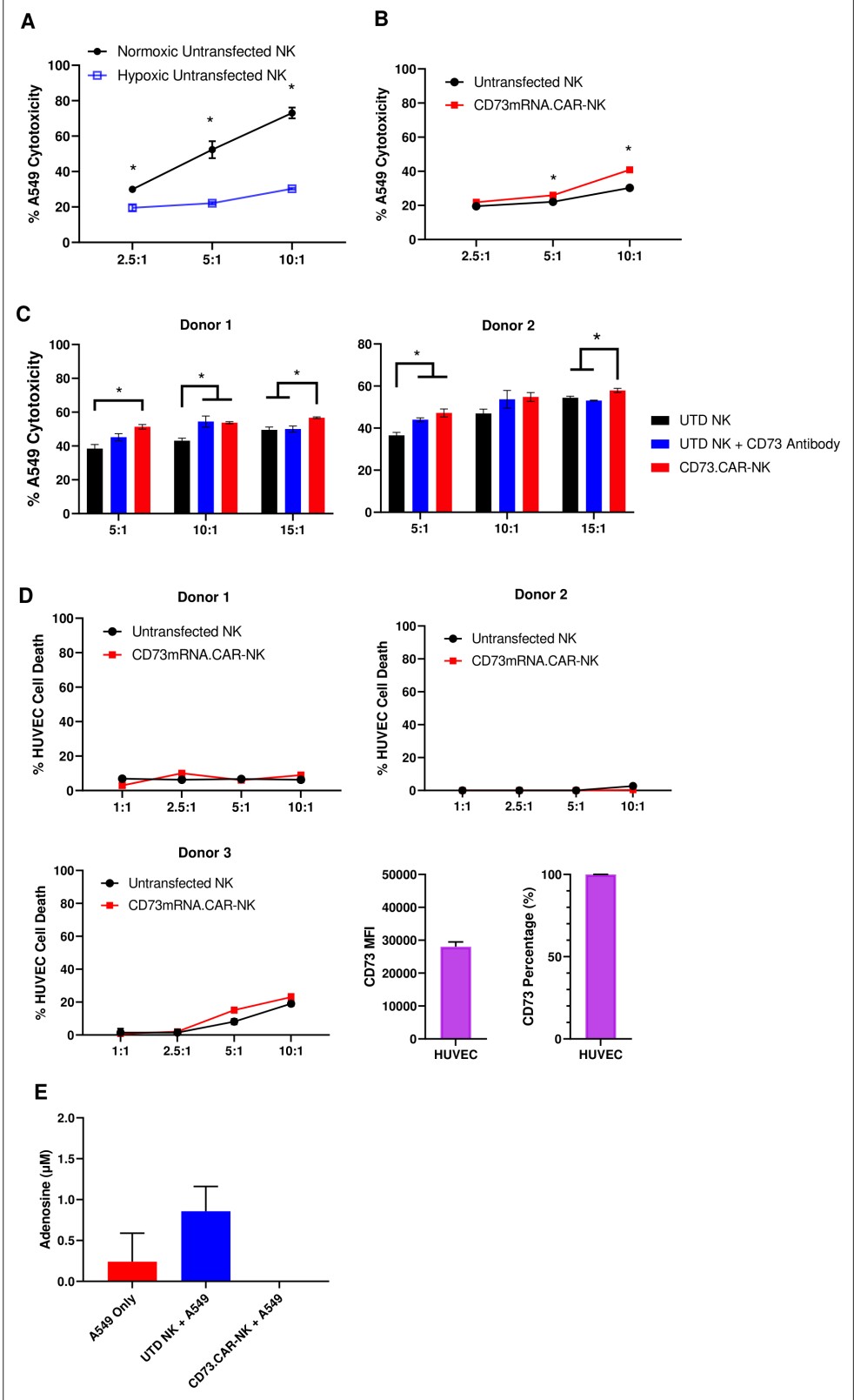

**Figure 4.** CD73.CAR NK cells rescue NK cell dysfunction under hypoxia. (**A**) Cytotoxicity of human primary NK cells against lung adenocarcinoma targets under normoxic and chemically-induced hypoxic conditions. (**B**) Cytotoxicity of CD73.mRNACAR-NK cells and untransfected primary human NK cells against lung adenocarcinoma targets under chemically-induced hypoxic conditions. (**C**) Killing of lung adenocarcinoma cells by CD73.mRNACAR-NK

*Figure 4 continued on next page*

*Figure 4 continued*

cells and anti-CD73 antibody (7G2) for two individual donors. (**D**) Killing of HUVEC cells by CD73.mRNACAR-NK cells and primary human NK cells for three individual donors. (**E**) Production of adenosine by lung adenocarcinoma cells after treatment with CD73.CAR-NK cells and untransduced primary human NK cells for three individual donors. Unless otherwise stated, data are representative of three donors. *p<0.05.

in tumor growth over time without any observable toxicities to normal cells. These studies are helping to contribute knowledge to immunotherapies that target the immunosuppression in solid tumors induced by adenosinergic metabolism driven by CD73.

## Discussion

In the lung cancer tumor microenvironment, the accumulation of adenosine is an exceedingly immunosuppressive mechanism that promotes the growth and survival of tumor cells while preventing immune cell function and cytotoxicity. This results in CD73 being a targetable receptor with prognostic value in lung cancer patients. The role of adenosinergic signaling in contributing to NK cell immunosuppression has long been recognized, *Chambers and Matosevic, 2019* and we have previously shown that the tumor microenvironment adenosine has resulted in the impaired metabolic, cytotoxic, and antitumor functions of NK cells *Chambers et al., 2018*. These high concentrations of adenosine that cause immunometabolic reprogramming occur from the tumor cells hypoxic and glycolytic fueling, which favor heightened activity of CD39 and CD73 on the cancer cells. The main function of CD39 and CD73 is to convert ATP to AMP and subsequently into adenosine to limit excess immune responses; however, cancer cells take advantage of this mechanism to prevent immune attack and promote tumor growth *Wang and Matosevic, 2018*; *Chambers and Matosevic, 2019*; *Allard et al., 2017*.

We evaluated the targeting of CD73 through the direct engagement of CAR-based signaling by engineering the NK cells to express a FCγRIIIa (CD16) based CAR directed at CD73. We used NK cell signaling domains based on CD16 to engineer the construct in order to promote NK cell cytotoxicity in the TME through an NK-specific CAR. Also, utilizing FCγRIIIa (CD16) taps into NK cells native ADCC (antibody-dependent cellular cytotoxicity) abilities *Yeap et al., 2016*. As a tumor antigen, CD73 has been shown to be a viable target, highly expressed in lung adenocarcinoma, with increased expression in mutated EGFR[+] lung adenocarcinoma patients *Velez and Burns, 2019*; *Streicher et al., 2017*. Additionally, lung adenocarcinoma also has high levels of adenosine accumulation *Blay et al., 1997*; *Ohta et al., 2006*.

Therapeutic interventions aimed at ablating activity of CD73 have so far favored antibody blockade, either alone or in combination with the targeting of other receptors involved in the adenosine signaling axis, such as A2A or A2B *Chambers and Matosevic, 2019*; *Perrot et al., 2019*; *Geoghegan et al., 2016*. Small molecule inhibitors of CD73 have also been used; however, these risk competing with extracellular AMP, a design concern that is eliminated with the use of antibodies *Geoghegan et al., 2016*; *Bowman et al., 2019*. Immunotherapy is another potential and viable treatment for lung adenocarcinoma *Qin et al., 2018*; *Zhang et al., 2019* and studies have shown that compared to antibody Fc, CARs can mediate stronger NK cell responses against tumor targets *Boissel et al., 2013*. In this study, we describe the first engineered NK therapy to target the immunosuppression caused by CD73 using a CAR-based therapy with NK cell signaling domains.

NK cells engineered with an anti-CD73 CAR effectively killed CD73[+] lung adenocarcinoma A549 cells when engineered with either mRNA or lentivirally transduced, showing that both transient and stable expression of the gene is enough to produce NK cell based anti-tumor effects. These effects are, however, limited in vitro because enzymatic activity of CD73, which drives the extent of adenosine accumulation, is typically substantially lower in vitro than in vivo. Moreover, CD73.CAR NK cells were healthy as they proliferated rapidly after transduction and stayed viable after mRNA electroporation. When comparing CD73-antibody to CD73.CAR-NK cells, cytotoxicity was lower for the CD73-antibody treatment, especially at the higher effector to target ratio in vitro, which may also correlate to antibody potency. Antibodies do not home into the sites of disease as efficiently as CAR-engineered cells and therefore, do not have as strong of responses in patients with large tumor burdens. Moreover, engagement of NK cells in the setting of antibody therapy relies on ADCC activity via CD16, which is

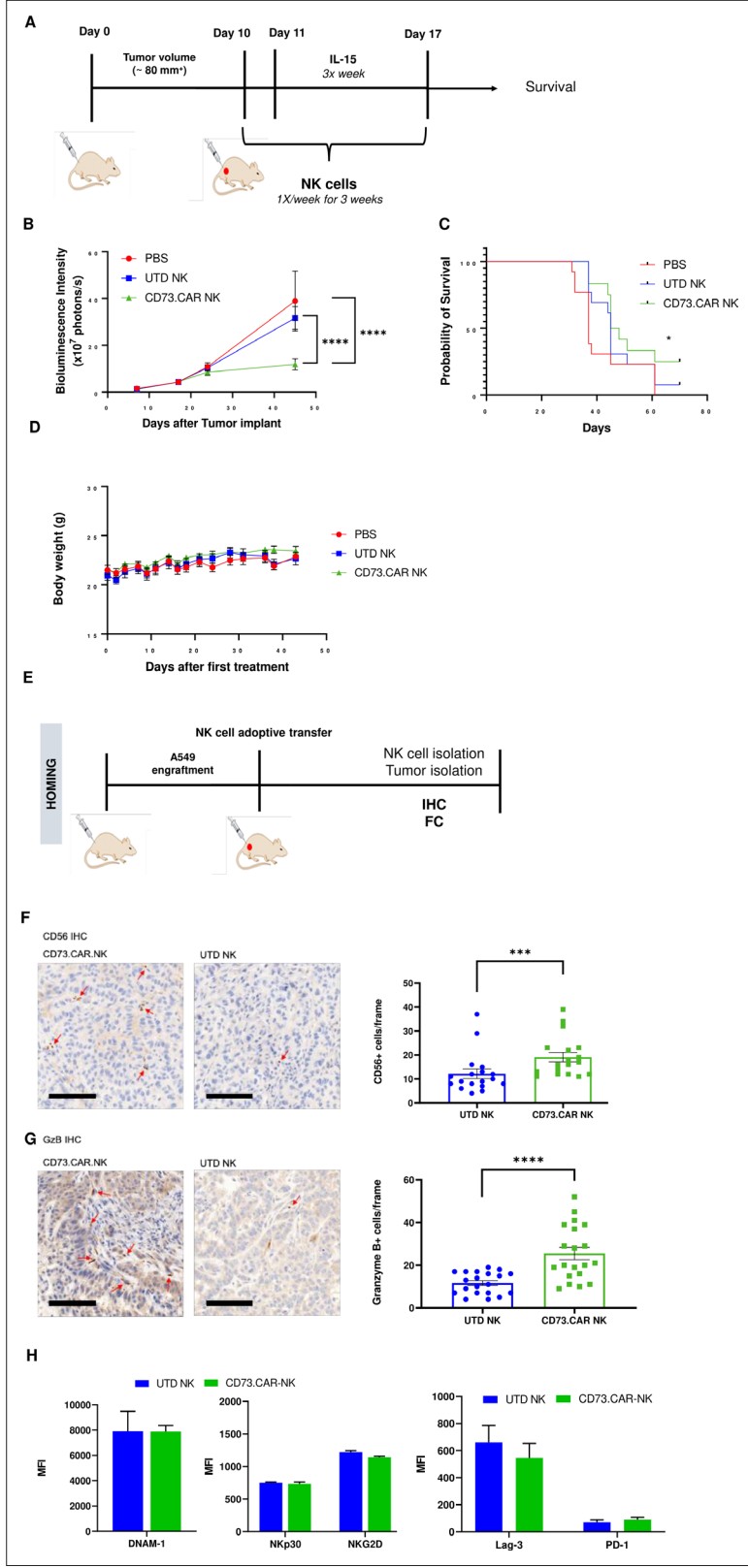

**Figure 5.** In vivo efficacy of CD73CAR NK cells in lung cancer xenografts. (**A**) Treatment schedule of A549 NSG xenografts with CD73.CAR-NK cells or experimental controls (n=6/group). (**B**) Tumor growth curve of A549 xenografts treated with CD73.CAR-NK cells, human NK cells or untreated control over 48 days. Statistical significance shown between CD73.CAR-NK cells and controls. (n=16) (**C**) Kaplan-Meier survival

*Figure 5 continued on next page*

*Figure 5 continued*

curve for A549 tumors treated with CD73.CAR-NK cells, human NK cells or untreated controls. Statistics shown between CD73.CAR-NK cells and untransduced controls. (**D**) Body weights of mice treated with CD73. CAR NK cells, human NK cells or untreated controls (n=16 mice per group). (**E**) General diagram for homing studies. (**F**) Immunohistochemical staining of treated A549 tumors for presence of NK cells (NKp46), and (**G**) immunohistochemical detection of granzyme B production in the same tumors. Scale bar = 100 µm; ×20 magnification. (**H**) Phenotyping of circulating NK cells isolated from treated A549-bearing NSG mice for activating markers DNAM-1, NKp30, and NKG2 and inhibitory markers LAG-3 and PD-1. UTD = untransduced, *p<0.05; ***p<0.001, ****p<0.0001.

often impaired in solid tumors. Studies have also shown that the biodistribution of an antibody treatment detected in solid tumors is a very small fraction of what is originally administered *Caruana et al., 2014*; *Goldenberg, 1988*. Redirecting the anti-CD73 NK cells also strengthened NK degranulation and cytokine release as evidenced by CD107a and IFNγ expression *Cohnen et al., 2013*.

Direct engagement of CD73 within a CAR construct not only mobilizes NK cell cytotoxicity, but promotes site-specific killing of CD73⁺ targets. Prior evidence also supports this approach, as a CD73 blockade *Antonioli et al., 2016*; *Geoghegan et al., 2016* was shown to result in tumor inhibition from recruitment of NK cells *Wang et al., 2018*, and an increase in the presence of IFN-γ and perforin *Young et al., 2016*; *Häusler et al., 2014*.

Under hypoxia, cancer cells express HIF-1α, which helps to increase adenosine accumulation, but NK cells have impaired killing ability and lowered expression of NKG2D and intracellular perforin and granzyme B (*Solocinski et al., 2020*). We also show decreased NK killing under hypoxic conditions; however, the CD73.CAR NK cells were able to rescue the detrimental effects of hypoxia to increase NK cell killing ability through the blocking of adenosine. NK cells also express HIF-1α under hypoxic conditions that cause this downregulation of killing ability. HIF-1α has been under investigation as an anti-tumor strategy but targeting HIF-1α directly in the NK cell could potentially be disadvantageous by impairing tumor killing and NK infiltration *Krzywinska et al., 2017*. Therefore, by targeting adenosine accumulation directly, the effects of hypoxia on the cancer cells will be diminished, while not affecting the killing ability of the NK cells.

One concern with the direct targeting of CD73 is the expression of this enzyme on healthy tissues which could increase the risk of on-target, off-site side effects. CD73 expression on non-tumor locations normally supports a variety of physiological processes to prevent an overactive immune system, and these locations include some epithelial cells, endothelial cells, smooth muscle cells, and cardiac myocytes *Kordaß et al., 2018*. Although no broad adverse safety effects have been reported in clinical studies targeting CD73 with systemically administered antibodies *Harvey et al., 2020*; *Siu et al., 2018*, a strategy to reduce the risk of targeting non-tumor tissue has been to engineer the antibodies without Fc receptor engagement. One such antibody, MEDI9447, is currently in clinical trials. Recent data from the MEDI9447 trial reported that systemic anti-CD73 therapy was well-tolerated with a manageable safety profile *Geoghegan et al., 2016*; *Overman et al., 2018*. However, eliminating the Fc engagement also eliminates the induction of ADCC, thus restricting NK cell activation and efficacy of treatment. Here we showed that inherently the NK cells did not have any observable off-target effects, as exhibited through killing experiments on endothelial cells. Moreover, the engineered NK cells did not have an increase in killing, indicating that even with high CD73, NK cells do not target normal healthy tissues, thus able to avoid off-target activation against non-malignant cells. While the use of HUVEC cells may have some limitations compared to normal endothelial cells, concerns over the targeting of CD73-directed NK cells to non-tumor tissues can be alleviated by administering these cells locally, such as intratumorally *Dubinett et al., 2010*; *Lee et al., 2017*.

Here, we also show that the engineered NK cells with the anti-CD73 construct stunted tumor growth and promoted increased NK cell filtration into lung tumors in vivo. These data point to CD73, as a tumor-associated antigen, being a viable target and mediating potent CAR-NK cell based antitumor effects against CD73⁺ lung adenocarcinoma xenografts in mice, both in terms of tumor metabolism and mouse survival. Clinical data has shown that adoptive transfer of cytokine-induced killer cells is well tolerated and provides improved outcomes over chemotherapy in treating NSCLC patients *Zhang et al., 2015*; *Wang et al., 2014*. To have the most effective treatment, we used multiple injections, with treatments once a week and IL-15 three times per week. NK cells can have a short lifespan,

but the addition of multiple injections and IL-15 helps to drive NK expansion and persistence *Liu et al., 2018*; *Sahm et al., 2012*; *Hoyos et al., 2010*. In our study, we employed three doses of the engineered NK cells and showed suppressed tumor growth after these three infusions. We did not have any evidence of toxicity of our treatment or with IL-15 as mice weights were consistent between treatment groups throughout the study. As evidenced by the smaller tumor size and stunted tumor growth over time after treatment ended, the NK cells were able to persist and maintain their cytotoxic potential. At the end of treatment, NK cells were still found in the circulation, and the engineered CD73.CAR-NK cells were found to have infiltrated into the tumors more than the non-engineered NK cells.

There were no differences between activation and inhibitory receptors in circulating NK cells for engineered and non-engineered groups; however, the granzyme B expression in the TME was much higher for the engineered NK cells. Since the treatment was injected via the tail vein, not all the cells in the circulation may have made it to the tumor site *Ma et al., 2019*; *Wang and Matosevic, 2020*. Moreover, the increase in granzyme B in the TME for the engineered cells showed that the NK cells maintained their cytotoxic potential over the course of the study.

Although engagement of CD73.CAR with CD73 on the tumor cells promoted NK cell-mediated cytotoxicity, we have not directly tested the activation signaling function of our CD73.CAR construct. Therefore, we cannot rule out whether this effect is dependent upon activation signaling through the CD16 domains or resulting from improved target cell conjugation to enhance natural cytotoxicity.

Therapeutically, NK cells have an advantage over the FDA approved T-cell therapies, as NK cells exert an innate killing ability that is non-CAR specific, as shown by the killing of tumor targets by non-engineered NK cells. Directing NK cells against CD73 in the TME can also potentially eliminate CD73 expressing suppressor cells, such as CD73$^+$ Tregs *Chambers and Matosevic, 2019*; *Schuler et al., 2014*. Expression of CD73 on tumor-infiltrating NK cells is low (*Chambers et al., 2022*), as we and others have shown, but CD73$^+$ NK cells were associated with a hyper-functional phenotype, including the ability to produce IL-10 (*Neo et al., 2020*). Therefore, targeting CD73 expressing cells in the TME will not only eliminate tumor-positive cells, but may remove these additional cell populations that help support the tumor niche. In conclusion, by using NK-specific signaling components to arm the NK cells against the immunosuppression in the TME, we have demonstrated a promising new immunotherapeutic modality to improve the survival of NSCLC patients and other patients with CD73$^+$ solid tumors.

## Materials and methods
### Materials availability

All data generated or analyzed during this study are included in the manuscript and supporting files, or are available at https://doi.org/10.5061/dryad.931zcrjnp. Source data have been provided for *Figure 1—source data 1*, *Figure 1—source data 2* (*Tables S1 and S2*). All reagents will be made available on request after completion of a Materials Transfer Agreement.

### Human peripheral blood NK cells and cell lines

Using negative selection, NK cells were isolated from whole blood with the EasySep Direct Human NK Cell Isolation Kit (StemCell Technologies). The gating strategy for isolation of CD56$^+$CD3$^-$ NK cells is shown in *Figure S1* (*Figure 2—figure supplement 1*). After isolation, NK cells were stimulated with mitomycin C (Cayman Chemical) treated K562 cells at a 1:1 or 1:2 (NK cell: feeder cell) ratio. Cells were placed in culture with RPMI (Gibco) with 10% FBS (Corning), 1% penicillin/streptomycin, 500 IU/mL IL-2 (Akron Biotech), 50 ng/mL hIL-21 (Gold Bio), and 50 ng/mL 41BBL (Peprotech) (RMPI feeder media) starting on day 0. Cells were cultured at a starting density of 5 × 10$^5$ cells/mL primary NK cells.

Commercial cell lines used were K562, A549, HEK293T, and HUVEC (ATCC) cells. Identity has been authenticated by STR profiling, and all cell lines tested negative for mycoplasma. K562 was maintained in IMDM (Gibco) supplemented with 10% FBS (Corning) and 1% penicillin/streptomycin (Gibco). A549 and HEK293T cells were cultured in DMEM (Gibco) supplemented with 10% FBS and 1% penicillin/streptomycin. HUVEC cells were cultured on a 1% gelatin coated surface in EGM-2 Endothelial Cell Growth Medium-2 BulletKit (Lonza) following the manufacturer's instructions with an

additional 6% FBS. To make feeder cells, K562 cells were chemically treated with 50 ug/mL mitomycin C (Cayman Chemical) for three hours.

## Plasmid construction and lentivirus production

The CD73-Fc(gamma)RIII_pcDNA3.1(+) plasmid was custom-cloned by Genscript using the mammalian expression vector pcDNA3.1(+) plasmid with the T7 promoter. The construct consisted of an anti-CD73 scFv derived from MEDI9447, which is an ScFv targeted antibody currently in clinical trials that binds to CD73 to block ectonucleotidase activity *Lonberg, 2016*, and a transmembrane and intracellular domain derived from FcγRIIIa (CD16). To generate the CAR structure, sequences corresponding to the truncated extracellular domain of Fc(gamma)RIII fused to the full intracellular and cytoplasmic domains of the Fc(gamma)RIII (Uniprot P08637; *FCGRA*) were combined with the anti-CD73 scFv and a mammalian signal sequence.

The pLV[Exp]-Puro-EF1A>{CD73} plasmid was custom-cloned by VectorBuilder using the mammalian gene expression lentiviral vector pLV, expressed under the EF1A promoter. The construct was similar to the first plasmid, with the MEDI9447 anti-CD73 scFv and FCγRIIIa (CD16) components. To produce lentiviral particles, 293T cells were transfected with the lentiviral VectorBuilder plasmid, VSVG envelope plasmid (PMD2.G, Addgene), and HIV-1-gag-pol helper plasmid (pspax2, Addgene) using Lipofectamine 3000 according to the manufacturer's instructions. Supernatant containing the lentivirus particles was collected at 48 and 72 hr and was concentrated using ultracentrifugation at 24,500 RPM for 2 hr at 4 °C using the 32TI swinging bucket rotor. The lentiviral titer was measured using the qPCR Lentivirus Titer Kit (abm) or the p24 ELISA following the manufacturer's instructions.

## mRNA transcription

mRNA from the CD73-Fc(gamma)RIII_pcDNA3.1(+) plasmid was first linearized using MluI and XbaI restriction enzymes. Then, the mRNA was synthesized following the HiScribe T7 ARCA mRNA Kit (with tailing) (New England Biolabs). Once the mRNA was synthesized, it was purified and concentrated using the NucleoSpin RNA Clean-up Kit (Macherey-Nagel). The concentration of mRNA was measured with the Qubit RNA BR Assay Kit and then stored at –80 °C in MilliQ water until use.

## Generation of engineered primary NK cells via mRNA electroporation and lentiviral transduction

For electroporated NK cells using the CD73-Fc(gamma)RIII_pcDNA3.1(+) plasmid, NK cells after isolation were first placed with mitomycin C treated K562 cells at a 1:2 (PNK:K562) ratio for at least two weeks. Then, NK cells were washed once in 1 x PBS and placed into Opti-MEM reduced serum media and into 4 mm electroporation cuvettes (Thermo Fisher) with 5 ug mRNA/1 million NK cells. NK cells were then electroporated at 300 V, 4ms pulse length, 5 pulses, and 5 second pulse interval with the BioRad Gene Pulser. After electroporation, cells were placed in pre-warmed RPMI feeder media and left to rest overnight.

For transduction using the lentiviral plasmid, pLV[Exp]-Puro-EF1A>{CD73}, peripheral-blood NK cells were activated with mitomycin C treated K562 cells at a 1:1 ratio (PNK:K562) for 5 days after isolation. After 5 days, NK cells were centrifuged and resuspended in media with RPMI +500 IU/mL IL-2 and 50 ng/mL IL-21. Viral supernatant was added to the cells at 30 MOI with 10 µg/mL protamine sulfate. Cells were then spun at 1000 G for 1 hr and incubated at 37 °C, 5% CO$_2$. About 6–8 hr after transduction, 10% FBS and 1.5 ng/mL IL-12 were added to the culture. The transduction process was repeated 24 hr later for a second round of transduction, and 24 hr after the last round of transduction, the cells were centrifuged and plated in fresh media to remove protamine sulfate. Three days after the final round of transduction, cells were stimulated again with mitomycin C treated K562 cells at a 1:2 ratio with RPMI feeder media + 1.5 ng/mL IL-12 for about two weeks to expand the cells before use.

## Gene expression by engineered NK cells

To measure the gene expression on both mRNA electroporated primary NK cells and lentiviral transduced primary NK cells, the cells were first collected and then washed once in 1 X PBS + 1% BSA. The cells were then incubated with 6 ng/mL CD73, Avi-His-Tag biotin labeled recombinant protein (BPS Bioscience) for 30 min at 37 °C. After the first incubation, cells were washed and stained with surface markers, CD56 (PE, clone: CMSSB), CD3 (PE-CY7, clone:UCHT1), and Strep-BV421 (Biolegend) and

incubated for 30 min at 4 °C. The cells were then washed and the Sytox green dead cell stain (Thermo Fisher) was added for cell viability. The gene expression was then detected via flow cytometry. Cells were also sorted by staining the cells with an anti-CD73 avi-His-Tag biotin labeled antibody and Strep-BV421 antibody at the Purdue Flow Cytometry Core to measure gene expression both before and after sorting.

## Viability of mRNA-electroporated NK cells

The viability of the primary NK cells after electroporation using the CD73-Fc(gamma)RIII_pcD3.1 + plasmid was determined using CCK-8 (ApexBio). NK cells were electroporated as described and then 100,000 cells/well were seeded in a 96-well plate at 24 hr, 48 hr, and 72 hr after electroporation. CCK-8 reagent was added to the wells for four hours and then cell viability was measured using a 450 nm wavelength.

## CRISPR knockout of lung cancer cells

To generate a knockout of CD73 on A549 cells, the Gene Knockout Kit v2-NT5E (Synthego) was used according to the manufacturer's protocol. Briefly, the sgRNA and cas9 were mixed with the Lipofect-amine CRISPRMAXreen Dead Cell Stain (Thermo Fisher) before running on flow cytom Cas9 Transfection Reagent (Thermo Fisher) following the manufacturer's instructions. The complex was incubated at room temperature for 10 min and then mixed with the A549 cells before plating in a 24-well plate and placed in the 37 °C, 5% $CO_2$ for 2–3 days. Cells were sorted at the Purdue Flow Cytometry Core to obtain 100% CD73 knockout A549 cells.

## Functional in vitro killing assay

To determine the cytotoxic killing ability of the engineered NK cells against tumor targets, NK cells were co-cultured with different target cells (A549, A549.CD73KO, and HUVEC) at E:T (NK cells:cancer cells) of 1:1, 2.5:1, 5:1, and 10:1 for 4 hr. The killing ability of the NK cells was compared between engineered NK cells, non-engineered NK cells, and non-engineered NK cells with 10 µg/mL CD73 Antibody (7G2, Thermo Fisher). The CD73 antibody was added at the beginning of the assay with the NK cells. Killing ability of the NK cells was measured using the lactate dehydrogenase (LDH) cytotoxicity kit (GBiosciences) following the manufacturer's instructions.

For chemically-induced hypoxic experiments, A549 cells were attached to the plate for at least four hours before co-incubation. One hour before co-incubation, cancer cell media was replaced with NK cell culture media containing 250 µM cobalt chloride and the cancer cells were left to rest in the incubator at 37 °C, 5% $CO_2$. NK cells were then added into the wells at the 2.5:1, 5:1, and 10:1 E:T ratios for 4 hr before measuring the killing with the LDH cytotoxicity kit.

## Functional degranulation and cytokine production

To test for degranulation via CD107a and for cytokine production via IFN-γ, CD73.Fc(gamma)RIIIa-PNK engineered cells and non-engineered cells were co-cultured with A549 at effector:target (PNK:A549) ratios of 2.5:1, 5:1, and 10:1 for 4 hr. Before the co-incubation, CD107a (APC, Clone: H4A3) was added to the culture. After 1 hr of co-incubation, monensin (Biolegend) and brefeldin A (Biolegend) were added to the cultures. Then the cells were collected, washed with FACS buffer (1 X PBS, 5% FBS), and stained for 30 min at 4 °C with recommended amounts of antibodies, CD56 (PE, clone: CMSSB) and CD3 (PE-CY7, clone:UCHT1). The Live/Dead fixable violet stain (Thermo Fisher) was also added for cell viability. Then cells were fixed and permeabilized with the Fixation/Permeation kit (BD Biosciences) and stained with IFNγ (PerCP/Cy5.5, clone: 4D.B3). After staining, cells were washed with FACS buffer and detected by flow cytometry. Gating was performed using both dead cell staining and CD3⁻CD56⁺ cells.

## Bioinformatics analysis

Lung cancer RNA-seq data (533 tumor and 59 normal samples) were downloaded from TCGA through the Genomic Data Commons using the TCGA-Biolinks package *Colaprico et al., 2016*. Next, the lung cancer patients (N=533) were classified into high/low groups based on expression of *NT5E* using the upper and lower cutoffs for high (50%) and low (50%) expression, respectively. Gene set enrichment analyses (GSEA) *Subramanian et al., 2005*; *Mootha et al., 2003* were carried out using KEGG, GO

Biological Processes and Immunological Signature databases. Additionally, a custom NK gene set was used in performing a GSEA, comprised of five genes (*NCR1, NCR3, KLRB1, CD160, PRF1*).

Survival analysis based on expression of *NT5E* was done in Kaplan-Meier Plotter using TCGA gene expression data. Data was analyzed and generated using a Kaplan–Meier curve for overall survival (OS). Kaplan–Meier curves were generated with a median or quartile survival cutoff. The estimation of hazard ratios (HR) was done by Cox proportional hazards model regression analysis. A 95% confidence interval was set and used. Patient samples with expression level above the threshold were considered as the high-expression or high-risk cohort.

Analysis of NSCLC [(TCGA, Firehose Legacy; PMID: 22588877)–230 patients with RNA-seq data] and SCLC [Small Cell Lung Cancer (U Cologne, Nature 2015; PMID:23550210)] – 81 patients with RNA-seq data study was performed on cBioPortal.

## Mice
Female 6- to 8-week-old NOD-*Rag1*[-/-]*IL2rg*[-/-], NOD rag gamma (NRG) mice were maintained at the Purdue Center for Cancer Research. All animal experiments were approved by the Purdue University Animal Care and Use Committee (protocol 1112000342).

## In vivo xenograft studies
A549 cells were first genetically manipulated to express firefly luciferase (A549-Luc) in order to monitor the tumor growth through in vivo bioluminescence imaging. The firefly luciferase labeled A549 cells were generated using a luciferase-based lentiviral vector according to the manufacturer's protocol (Puro Lentiviral Particles, LVP325).

A549 cells ($2 \times 10^6$) cells were inoculated subcutaneously (SC) in the right flank of NRG mice (n=6–7/group) with 100 µL matrigel. PBS (100 µL) and engineered NK and non-engineered NK cells at 5 million cells/mouse were injected once a week via tail vein after the tumors were approximately 80 mm³. Also, 1 µg/mouse of IL-15 (Shenandoah Biotech) was injected three times per week with an intraperitoneal injection (ip). Tumor size was measured using the Spectral Ami Optical imaging system once per week. Mice were injected with 10 µL/g and 150 mg Luciferin/kg body weight with a stock solution of 15 mg/mL D-Luciferin (Syd Labs). The D-Luciferin was injected intra-peritoneally (i.p.) for 10 min before imaging. Body weights of the mice were also recorded during the treatment period. Tumor growth was monitored until mice met predefined endpoint criteria. Tumor tissues were processed in 10% neutral-buffered formalin for histological IHC analysis. For homing studies, NK cells were administered once per week with IL-2 three times per week, and NK cells were detected by IHC as described.

## Immunohistochemical analysis
Immunohistochemistry (IHC) staining was performed at the Histology Research Laboratory at the Purdue University College of Veterinary Medicine. Mice tumors were fixed in 10% neutral-buffered formalin for 24 hr before being transferred to 70% ethanol, embedded in paraffin, and cut into 3–5 µm sections. The NK cells in the tumors were stained using the CD56 monoclonal antibody (clone 56C04, ThermoFisher) at a 1:200 dilution, granzyme B using the Granzyme B antibody (Abcam, ab4059) and CD73 using an anti-rabbit CD73 antibody (Cell Signaling Technologies, D7F9A). Images were analyzed using FIJI (ImageJ) through color deconvolution, setting a threshold, creating a binary image, and analyzing the number of particles present. Intensity of the images was also observed using the mean gray value and optical density.

## Immunohistochemistry of lung cancer PDX
LSL-KrasG12D/Tp53flox/flox mice were set up and maintained following established procedures *Capaccione et al., 2014*; *Jackson et al., 2005*. Mice were infected by intratracheal instillation adenovirus expressing Cre at $4 \times 10^7$ units per mouse at 6–8 week of age. Presence of tumors was confirmed 16 weeks after AdenoCre inhalation by low-resolution CT images on a Siemens Inveon PET/CT. Mice were euthanized at 22 weeks post Adeno-Cre and tumors collected for further analysis. Separately, autologous lung adenocarcinoma (LUAD) cells derived from the K-ras[LSL-G12D/+];p53[fl/fl] (KP) model of LUAD were engrafted into C57BL/6 mice. On day 31, tumors were collected for imaging analysis. Immunohistochemistry to detect CD73 expression was done as described above.

## Phenotypic analysis of circulating NK cells

Mouse blood was collected from anesthetized mice using cardiac puncture. Mouse blood from each group (engineered and non-engineered NK) was pooled into heparin or EDTA coated tubes. Blood was then removed from each tube and resuspended in ACK lysis buffer (Thermo Fisher) following the manufacturer's instructions to lyse the blood cells. After the majority of the blood cells were removed, the cells were washed twice in 1 X FACS buffer and then stained with CD56 (PE-Cy5.5, clone CMSSB), CD3 (PE-CY7, Clone: UCHT1), LAG-3 (BV-421, clone: 11C3C65), PD-1 (APC, clone: EH12.2H7), DNAM-1 (PE, clone: Dx11), NKp30 (BV711, clone: P30-15), or NKG2D (APC, clone: 1D11) for 30 min at 4 °C. Cells were then washed and stained with Sytox Green Dead Cell Stain (Thermo Fisher) before running on flow cytometry.

## Adenosine production

To test for CD73 activity and ability of the A549 cells to generate adenosine after treatment with the lentiviral engineered NK cells, A549 cells were plated similar to the functional degranulation and cytokine production assays with a 10:1 E:T ratio and $2 \times 10^5$ PNK cells/sample. After two hr of co-incubation, adenosine monophosphate (AMP) at 250 µM was added into the cell culture media and incubated for 30 min at 37 °C. Then the cell culture supernatants were collected, and the adenosine concentrations were determined following the Adenosine Assay Kit.

## Statistical analysis

Prism 9 (Graphpad Software) was used for all statistical analysis with a $p < 0.05$ (*) as significant. Data was presented as mean +/- standard error of the mean (SEM), normality of the data was tested with the Shapiro Wilk test, and equal variances was checked with the Brown-Forsythe test. Ordinary one-way analysis-of-variance tests were used for multiple-group comparisons, and differences between two groups were evaluated using the two-tailed student's t-test. For the animal study, linear regression was used to assess the tumor growth curve and Kaplan-Meier survival was used for survival analysis.

## Acknowledgements

This work was supported by the V Foundation for Cancer Research (Grant #D2019-039), the Walther Cancer Foundation (Embedding Tier I/II Grant #0186.01), a Migliaccio/Pfizer Graduate Fellowship and a Purdue Research Foundation Graduate Fellowship to Andrea Chambers, and a McKeehan Graduate Fellowship to Kyle Lupo. The authors also gratefully acknowledge the support of the Collaborative Core for Cancer Bioinformatics, the Biological Evaluation Shared Resource and the Flow Cytometry Shared Resource, with support from the Purdue Center for Cancer Research, NIH grant P30 CA023168, the IU Simon Cancer Center NIH grant P30 CA082709, and the Walther Cancer Foundation.

## Additional information

### Funding

| Funder | Grant reference number | Author |
|---|---|---|
| V Foundation for Cancer Research | #D2019-039 | Sandro Matosevic |
| Lilly Graduate Fellowship | 2019 | Andrea M Chambers |
| Walther Cancer Foundation | 0186.01 | Sandro Matosevic |
| Migliaccio/Pfizer Graduate Fellowship | 2019-2020 | Andrea M Chambers |
| National Cancer Institute | P30 CA023168 | Sagar Utturkar Nadia Lanman Sandra Toregrosa-Allen Bennett D Elzey |

| Funder | Grant reference number | Author |
|---|---|---|
| National Cancer Institute | P30 CA082709 | Shadia Jalal |

The funders had no role in study design, data collection and interpretation, or the decision to submit the work for publication.

## Author contributions
Andrea M Chambers, Conceptualization, Formal analysis, Investigation, Methodology, Writing – original draft, Writing – review and editing; Kyle B Lupo, Jiao Wang, Jingming Cao, Investigation; Sagar Utturkar, Nadia Lanman, Resources, Software, Investigation, Methodology; Victor Bernal-Crespo, Shadia Jalal, Sandra Torregrosa-Allen, Bennett D Elzey, Resources, Investigation, Methodology; Sharon R Pine, Resources, Investigation; Sandro Matosevic, Conceptualization, Resources, Formal analysis, Supervision, Funding acquisition, Investigation, Methodology, Writing – original draft, Project administration, Writing – review and editing

## Author ORCIDs
Sandro Matosevic ![ORCID] http://orcid.org/0000-0001-5118-2455

## Ethics
Ethics statementPeripheral blood NK cells were obtained from 15 normal healthy donor volunteers who gave written consent through Purdue University's IRB protocol (#1804020540). All procedures performed were approved by Purdue University's Institutional Review Board (IRB).

## Decision letter and Author response
Decision letter https://doi.org/10.7554/eLife.73699.sa1
Author response https://doi.org/10.7554/eLife.73699.sa2

# Additional files

## Supplementary files
• Transparent reporting form

## Data availability
All data generated or analyzed during this study are included in the manuscript and supporting files, or are available at https://doi.org/10.5061/dryad.931zcrjnp. Source data have been provided in Figure 1—source data 1 and 2.

The following dataset was generated:

| Author(s) | Year | Dataset title | Dataset URL | Database and Identifier |
|---|---|---|---|---|
| Matosevic S | 2022 | Data from: Engineered natural killer cells impede the immunometabolic CD73-adenosine axis in solid tumors | https://doi.org/10.5061/dryad.931zcrjnp | Dryad Digital Repository, 10.5061/dryad.931zcrjnp |

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
