## [Editor Report]

The adoptive transfer of NK cells expressing chimeric antigen receptors (CARs) has generated significant interest for cancer immunotherapy, considering recent success of CAR-NK cells in treating B cell malignancies with fewer deleterious side effects, as compared to CAR-T cell therapies. In this paper, the authors engineered NK cells expressing a CAR targeting CD73, which is highly expressed in lung adenocarcinomas and generates immunosuppressive adenosine. Their CD73.CAR-NK cells not only exhibited improved infiltration of and enhanced cytotoxicity toward CD73-expressing tumor cells, but also significantly reduced adenosine production by a lung adenocarcinoma line, which has potential to further improve anti-tumor immune responses.

---

## [Decision Letter]

**Decision letter after peer review:**

Thank you for submitting your article "Engineered natural killer cells impede the immunometabolic CD73-adenosine axis in solid tumors" for consideration by *eLife*. Your article has been reviewed by 3 peer reviewers, including Kerry Campbell as the Reviewing Editor and Reviewer #1, and the evaluation has been overseen by Wafik El-Deiry as the Senior Editor.

Essential revisions:

1) Numbers of mice analyzed should be increased for experiments in Figure 5 as requested by reviewer #1.

2) The authors should provide some evidence whether or not the CAR is providing a direct activation signal into the NK cells upon engagement with CD73.

3) The authors should test whether CD73 is upregulated when the NK cells are activated. This may be difficult if the CAR blocks CD73 protein detection, but perhaps at least testing for up regulation of mRNA by RT-PCR could be performed.

4) Data from assay of untransfected NK cells should be added to Figure 4C.

*Reviewer #1 (Recommendations for the authors):*

1. With the minimal enhanced killing of A549 target cells in vitro, it is unclear if the CAR is actually associating with zeta and γ adaptors and thereby functional to signal into NK cells or just inhibiting CD73 catalytic activity on the target cells. Differences in adenosine production could be influencing natural cytotoxicity responsiveness in figures 1C&D, so the authors should treat CAR-bearing NK cells with biotinylated recombinant CD73 and streptavidin and test for degranulation and/or calcium mobilization. Ideally, they could also demonstrate whether the CAR physically associates with zeta and or γ adaptors in co-immunoprecipitation experiments.

2. Analysis of only 4 mice per group in the survival study of Figure 5B is grossly inadequate, and this should be increased to 15-20/group to achieve valid statistical rigor.

3. NK cell infiltration appears to be greater for the mice treated with CD73.CAR.NK in the single example shown in Figures 5G&H, which is particularly interesting, but confirmation of this potentially key point by quantification with mean and SD from at least 5 mice should be shown to significantly strengthen this conclusion and the manuscript overall. Was the decrease in Lag-3 in Figure 5I statistically significant? If not, more mice should be assayed before claiming Lag-3 expression was decreased. The number of mice assessed is not indicated. Also, Figures 5B and 5C are reversed, as are 5D and 5E. Also, descriptions of 5G,H, and I are mixed up in text and figure legend.

*Reviewer #2 (Recommendations for the authors):*

The authors should double check (and correct/modify as needed) some instances in the introduction. For example, where it is mentioned about non-targetable genotypes – here, the authors give KRAS mutant tumors as an example, but some KRAS mutants can be targeted, so this should be addressed. The authors make the broad statement that CAR T cells can induce GvHD, but they make no distinction between autologous and allogeneic CAR T cells. This should be more precisely written.

The methods the authors used appear to be suitable to address their main questions. There are some minor typos that should be corrected (e.g. ScFv is usually written as scFv, generally spaces should be given between numbers and units, "increase alter the killing" – should be either increase or alter, so one of the words should be deleted).

On page 17 of the Results section, the authors write that "overall high yields of expression of the CD73 CAR with no changes in NK cell phenotypic populations were obtained", but do not provide any link as to where these data are displayed. These are important points and should be shown.

The font size of the text in figure 1 should be increased and the size of the figure panels in general should be increased so that the reader can more easily view the data.

In figure 2, the authors should provide information regarding the number of replicates for each panel instead of only at the end of the figure. The data shown in figure 2C is not the transduction efficiency as the authors describe it – the transduction efficiency would need to be calculated in the transduced culture prior to sorting. What is shown here is rather the enrichment of modified cells after sorting. The authors should define the right and left plots in figure 2E.

In figure 3, how do the authors explain their observations of increased CD107a and IFN-γ expression at lower effector-to-target (E:T) ratios versus higher E:T?

How many times were the experiments shown in figure 4 accomplished?

The data panels shown in figure 5 do not match how they are described in the text (page 20). Figure 5C shows bioluminescence data, not survival, 5D shows the body weight, 5E the tumor volume, 5G NKp46 expression, 5H granzyme B levels, and H is shown twice. The authors should change these to match the data shown.

Do the authors have any data comparing the "NK-specific" with "T cell" CAR constructs in this context?

*Reviewer #3 (Recommendations for the authors):*

Although the experiments were well performed, the manuscript could be improved by addressing the points below.

1. CD73 could be induced on NK cells upon engagement with tumor cells, thereby impairing the antitumor effect of these engineered anti-CD73 CAR-NK cells in the TME. Thus, the authors should examine this possibility in their work.

2. The authors need to discuss why anti-CD73 CAR-NK cells had a minimal killing activity on CD73+ HUVEC cells compared to CD73+ human tumor cells.

3. In Figure 4C, the control group "untransfected NK cells only" is missing.

4. The tumor weight data (Figure 5E) about the antitumor effect of anti-CD73 CAR-NK cells vs untransduced NK cells seem inconsistent with the survival data (Figure 5B) as well as the bioluminescence data (Figure 5C). Are there any significant differences of tumor weights among different groups?

5. The name of the tumor should be "A549" rather than "GBM43" in Figure 5F.

6. Overall, please indicate the reproducibility of your animal and other important studies.

---

## [Author Response]

Reviewer #1 (Recommendations for the authors):1. With the minimal enhanced killing of A549 target cells in vitro, it is unclear if the CAR is actually associating with zeta and γ adaptors and thereby functional to signal into NK cells or just inhibiting CD73 catalytic activity on the target cells. Differences in adenosine production could be influencing natural cytotoxicity responsiveness in figures 1C&D, so the authors should treat CAR-bearing NK cells with biotinylated recombinant CD73 and streptavidin and test for degranulation and/or calcium mobilization. Ideally, they could also demonstrate whether the CAR physically associates with zeta and or γ adaptors in co-immunoprecipitation experiments.

We appreciate this thoughtful and important comment. To demonstrate that signaling along the CAR domains occurs upon CD73 engagement, we have carried out a cytotoxicity study using CRISPR/Cas9 CD73 KO A549 cells, which do not express CD73 and produce no adenosine *(Figure 3B)*. CAR-expressing NK cells did not show enhanced cytotoxicity against CD73 KO A549 cells, but did show enhanced cytotoxicity to A549 WT cells, indicating that CAR-bearing NK cells require CD73 engagement for functional activation.

Further, we have added new data which indicate that the killing ability of the CD73.CAR-

NK cells is higher against target cells than WT human NK cells in combination with antiCD73 antibody, suggesting the enhanced cytotoxicity of CAR-bearing NK cells is through CAR engagement rather than merely enzymatic blockade of CD73.

Though adenosine does affect NK cell activity, in a published manuscript, we showed that among these activity-supporting mechanisms, activation of NK cells in the presence of cytokines is able to temper adenosine-mediated suppression of NK cell effector responses, resulting in promotion of NK cell killing. In addition, the metabolic breakdown of adenosine into inosine contributes to its limited accumulation in vitro. See Figure 3B, 4C.

2. Analysis of only 4 mice per group in the survival study of Figure 5B is grossly inadequate, and this should be increased to 15-20/group to achieve valid statistical rigor.

We appreciate the feedback that the number of mice used in our original submission could be improved for statistical rigor. To address this, we have carried out a brand new in vivo efficacy study with a significantly larger number of mice. With the new study, 16 mice/group were added to achieve statistical rigor. Our new data are included in the manuscript, where we discuss our observations. We observed similar trends in terms of tumor growth and efficacy with a larger sample size and have included this updated data in the manuscript. See Figure 5.

3. NK cell infiltration appears to be greater for the mice treated with CD73.CAR.NK in the single example shown in Figures 5G&H, which is particularly interesting, but confirmation of this potentially key point by quantification with mean and SD from at least 5 mice should be shown to significantly strengthen this conclusion and the manuscript overall. Was the decrease in Lag-3 in Figure 5I statistically significant? If not, more mice should be assayed before claiming Lag-3 expression was decreased. The number of mice assessed is not indicated. Also, Figures 5B and 5C are reversed, as are 5D and 5E. Also, descriptions of 5G,H, and I are mixed up in text and figure legend.

We have carried out new IHC studies with 5 mice per IHC analysis group and have quantified infiltrating NK cells in each group. These data have been included in the manuscript. We have also updated the claims regarding LAG-3 expression to more accurately clarify that no difference in LAG-3 expression was observed and have corrected the order of the figures in the manuscript. See Figure 5, See Lines 341-346.

Reviewer #2 (Recommendations for the authors):The authors should double check (and correct/modify as needed) some instances in the introduction. For example, where it is mentioned about non-targetable genotypes – here, the authors give KRAS mutant tumors as an example, but some KRAS mutants can be targeted, so this should be addressed. The authors make the broad statement that CAR T cells can induce GvHD, but they make no distinction between autologous and allogeneic CAR T cells. This should be more precisely written.The methods the authors used appear to be suitable to address their main questions. There are some minor typos that should be corrected (e.g. ScFv is usually written as scFv, generally spaces should be given between numbers and units, "increase alter the killing" – should be either increase or alter, so one of the words should be deleted).

We appreciate these thorough observations. The above changes have been made to the text, including the introduction, to clarify and correct previous statements. scFv has also been replaced throughout the manuscript. We have also proofread and checked the entire manuscript for typos and writing errors and have made edits, where appropriate, throughout. See lines 59/60; line 79; line 174 and throughout manuscript

On page 17 of the Results section, the authors write that "overall high yields of expression of the CD73 CAR with no changes in NK cell phenotypic populations were obtained", but do not provide any link as to where these data are displayed. These are important points and should be shown.The font size of the text in figure 1 should be increased and the size of the figure panels in general should be increased so that the reader can more easily view the data.

The proportion of CD56^+^CD3^-^ NK cells before and after transduction is shown in Figure 2E. This has been included in the manuscript. We have clarified the section on gene expression and added citations to figures where missing and to improve clarity. See lines 143-158

The font size of the text in Figure 1 and other figures have been increased. See Figure 1 and rest of manuscript*.*

In figure 2, the authors should provide information regarding the number of replicates for each panel instead of only at the end of the figure. The data shown in figure 2C is not the transduction efficiency as the authors describe it – the transduction efficiency would need to be calculated in the transduced culture prior to sorting. What is shown here is rather the enrichment of modified cells after sorting. The authors should define the right and left plots in figure 2E.

We have made the clarification for the data shown as representing enrichment after cell sorting in the manuscript. We have also edited the transduction to reflect enrichment of CAR-NK cells. Replicates have been reported. See lines 143-146, lines 819-820, see Figure 2 caption.

In figure 3, how do the authors explain their observations of increased CD107a and IFN-γ expression at lower effector-to-target (E:T) ratios versus higher E:T?

We wish to thank the reviewer for this important comment, and for the opportunity to address it. We, and others, observe that CD107a and IFN-γ decrease with increases in effector:target ratio (Aarsund, et al., 2022; Gomez-Lomeli, et al. 2014). At lower E:T ratios there are more NK:cancer interactions per cell, and each individual NK cell responds to this heightened stimulus expressing more CD107a and IFNγ at lower E:T ratios, as measured by flow cytometry. Below we are including references which outline this is more detail.

– Aarsund, M., Segers, F.M., Wu, Y. *et al.* Comparison of characteristics and tumor targeting properties of extracellular vesicles derived from primary NK cells or NK-cell lines stimulated with IL-15 or IL-12/15/18. *Cancer Immunol Immunother* (2022). https://doi.org/10.1007/s00262-022-03161-0

– Gómez-Lomelí, P., Bravo-Cuellar, A., Hernández-Flores, G. *et al.* Increase of IFN-γ and TNF-γ production in CD107a + NK-92 cells co-cultured with cervical cancer cell lines pre-treated with the HO-1 inhibitor. *Cancer Cell Int* 14, 100 (2014). https://doi.org/10.1186/s12935-014-0100-1

How many times were the experiments shown in figure 4 accomplished?

Experiments were repeated in triplicate for three individual donors, which we have included in the figure caption. See Figure 4 caption.

The data panels shown in figure 5 do not match how they are described in the text (page 20). Figure 5C shows bioluminescence data, not survival, 5D shows the body weight, 5E the tumor volume, 5G NKp46 expression, 5H granzyme B levels, and H is shown twice. The authors should change these to match the data shown.

We appreciate the reviewer pointing these errors out. These have been corrected, and the text has been edited to match the updated figure with new data. See Figure 5F; lines 225-239.

Do the authors have any data comparing the "NK-specific" with "T cell" CAR constructs in this context?

We are thankful for this question. We have not carried out such comparison. Primarily because we intended to use the CD16 signaling domains only on NK cells (since they naturally signal within CD16, and optimization of such construct for T cells was beyond the resources available to us).

Reviewer #3 (Recommendations for the authors):Although the experiments were well performed, the manuscript could be improved by addressing the points below.1. CD73 could be induced on NK cells upon engagement with tumor cells, thereby impairing the antitumor effect of these engineered anti-CD73 CAR-NK cells in the TME. Thus, the authors should examine this possibility in their work.

We thank the reviewer for this important question, which we find very interesting. We have in fact examined the possibility of CD73 being induced on NK cells in response to cancer cells. We have analyzed patient data and have carried out a series of assays with both WT and CD73KO CD73^+^ cancer cells in the presence of NK cells both under normoxic and chemically-induced hypoxic conditions. Our data show that CD73 upregulation on NK cells is generally limited, and requires high expression of CD73 on cancer cells, as well as hypoxic conditions. In addition, NK cells isolated from tumors of patients show no significant upregulation of CD73 on their surface. This analysis highlights that while this phenomenon is possible, it results in limited upregulation of CD73 on the NK cell surface, thus limiting the loss of NK cells due to potential fratricide. These data have recently been accepted for publication in a separate manuscript (Chambers et al., *Cancer Immunology, Immunotherapy*, 2022, 10.1007/s00262-022-03219z).

2. The authors need to discuss why anti-CD73 CAR-NK cells had a minimal killing activity on CD73+ HUVEC cells compared to CD73+ human tumor cells.

We appreciate this important and throughtful question. Firstly, it has been shown that HUVECs have no effect on the proliferation or killing ability of NK cells (Pradier Cell Transplantation 2011). Additionally, others have reported low killing of HUVECs by NK cells at E:T ratios below 20:1 (Gorini et al., *Blood* 2010), and suggest that secretion of high levels of ATP may be protective of HUVECs from NK cell killing. They also express HLA-E, suggesting there could be MHC-dependent inhibition of NK cells, contributing to low killing. We are including the references showing this below. We have also edited the manuscript to address this.

– Gorini S. et al.; ATP secreted by endothelial cells blocks CX_3_CL1-elicited natural killer cell chemotaxis and cytotoxicity via P2Y_11_ receptor activation. *Blood* 2010; 116 (22): 4492–4500.

3. In Figure 4C, the control group "untransfected NK cells only" is missing.

We have repeated this assay to include untransfected NK cells only and have replaced the figure in the manuscript with an updated assay which also includes the untransfected NK cell group.

4. The tumor weight data (Figure 5E) about the antitumor effect of anti-CD73 CAR-NK cells vs untransduced NK cells seem inconsistent with the survival data (Figure 5B) as well as the bioluminescence data (Figure 5C). Are there any significant differences of tumor weights among different groups?

We appreciate this comment, as well as the opportunity to clarify this observation. Because mice were euthanized at a predetermined endpoint based on tumor size, at different timepoints, no significant differences in tumor weights were observed. In other words, no difference in tumor weight would be expected since tumors are sacrificed when the reach the same size. Bioluminescence data were collected at the same timepoints for all groups, and are a more valid metric for assessing the anti-tumor effect of anti-CD73 CAR-NK cells and untransduced NK cells. To address this and remove confusion, we have removed tumor weight measurements from the manuscript and have rather provided updated in vivo efficacy data with a significantly larger number of mice (16 per group). See figure 5.

5. The name of the tumor should be "A549" rather than "GBM43" in Figure 5F.

This has been corrected.

6. Overall, please indicate the reproducibility of your animal and other important studies.

We thank the reviewer for this question. Most in vitro and in vivo studies have been repeated at least three times and results are consistent between independent experiments. The number of replicates for specific experiments is provided in the manuscript. A new set of in vivo studies were repeated for this revision, with a total of 16 mice/group to strengthen the statistical significance of the data. NK cell transduction is typically dependent on donor, with average expression reported in Figure 2, and upon sorting, we consistently obtained a pure >99% population of CAR-expressing NK cells. Overall, we have been thorough in optimizing methodology to obtain and report consistent data and findings.